# Effect of Uphill Running on VO_2_, Heart Rate and Lactate Accumulation on Lower Body Positive Pressure Treadmills

**DOI:** 10.3390/sports9040051

**Published:** 2021-04-06

**Authors:** Daniel Fleckenstein, Olaf Ueberschär, Jan C. Wüstenfeld, Peter Rüdrich, Bernd Wolfarth

**Affiliations:** 1Institute of Sports Science, Humboldt University, 10099 Berlin, Germany; 2Department of Sports Medicine, Institute for Applied Training Science, 04109 Leipzig, Germany; wuestenfeld@iat.uni-leipzig.de (J.C.W.); ruedrich@iat.uni-leipzig.de (P.R.); 3Department of Engineering and Industrial Design, Magdeburg-Stendal University of Applied Sciences, 39114 Magdeburg, Germany; olaf.ueberschaer@h2.de; 4Department of Biomechanics, Institute for Applied Training Science, 04109 Leipzig, Germany; 5Department of Sports Medicine, Charité University School of Medicine, Humboldt University, 10117 Berlin, Germany

**Keywords:** AlterG, anti-gravity treadmill, body weight support, graded running

## Abstract

Lower body positive pressure treadmills (LBPPTs) as a strategy to reduce musculoskeletal load are becoming more common as part of sports conditioning, although the requisite physiological parameters are unclear. To elucidate their role, ten well-trained runners (30.2 ± 3.4 years; VO_2max_: 60.3 ± 4.2 mL kg^−1^ min^−1^) ran at 70% of their individual velocity at VO_2max_ (vVO_2max_) on a LBPPT at 80% body weight support (80% BW_Set_) and 90% body weight support (90% BW_Set_), at 0%, 2% and 7% incline. Oxygen consumption (VO_2_), heart rate (HR) and blood lactate accumulation (LA) were monitored. It was found that an increase in incline led to increased VO_2_ values of 6.8 ± 0.8 mL kg^−1^ min^−1^ (0% vs. 7%, *p* < 0.001) and 5.4 ± 0.8 mL kg^−1^ min^−1^ (2% vs. 7%, *p* < 0.001). Between 80% BW_Set_ and 90% BW_Set_, there were VO_2_ differences of 3.3 ± 0.2 mL kg^−1^ min^−1^ (*p* < 0.001). HR increased with incline by 12 ± 2 bpm (0% vs. 7%, *p* < 0.05) and 10 ± 2 bpm (2% vs. 7%, *p* < 0.05). From 80% BW_Set_ to 90% BW_Set_, HR increases of 6 ± 1 bpm (*p* < 0.001) were observed. Additionally, LA values showed differences of 0.10 ± 0.02 mmol l^−1^ between 80% BW_Set_ and 90% BW_Set_. Those results suggest that on a LBPPT, a 2% incline (at 70% vVO_2max_) is not yet sufficient to produce significant physiological changes in VO_2_, HR and LA—as opposed to running on conventional treadmills, where significant changes are measured. However, a 7% incline increases VO_2_ and HR significantly. Bringing together physiological and biomechanical factors from previous studies into this practical context, it appears that a 7% incline (at 80% BW_Set_) may be used to keep VO_2_ and HR load unchanged as compared to unsupported running, while biomechanical stress is substantially reduced.

## 1. Introduction

To improve physical performance and to avoid overuse injuries, elite runners are increasingly using alternative training tools such as the lower body positive pressure treadmill (LBPPT), an anti-gravity treadmill with an enclosed, air-filled chamber that generates positive pressure around the lower body [1]. Currently, it is known that running at the same velocity on LBPPT with partial body weight support is accompanied by a decrease in oxygen consumption (VO_2_) [1,2,3,4,5,6,7,8], heart rate (HR) [1,4,5,8] and blood lactate accumulation (LA) [2]—as compared to running under unsupported conditions. Furthermore, VO_2max_ and HR_max_ seem to be unaffected by LBPPTs [3,9].

The first systematic review on this topic was given by Farina et al., 2017 [4]. The authors concluded that running on LBPPTs is effective in reducing impacts (i.e., peak ground reaction forces). To still achieve the desired physiological stimuli (similar to those of an unsupported treadmill), faster running velocities and/or inclines need to be applied. Faster running velocities on the LBPPT are associated with an increase in physiological demand [1,2,3]. In turn, however, the increased velocity leads to higher ground reaction forces and loading rates [3,10,11,12,13]. Accordingly, velocity increases should be applied with caution. On the other hand, the factor *incline* could be suitable for increasing physiological demand while keeping biomechanical loads low. Recently, Farina et al. pointed out that there is a research gap on inclined running on a LBPPT, which “should be explored” [4] (p. 272). Especially because uphill running is an important training tool for middle and long distance runners [14].

Therefore, the purpose of the present study was to investigate the effects of LBPPT uphill running on VO_2_, HR and LA. Specifically, the aim was to determine the influence of three different inclines (0%, 2% and 7%) and of two different body weight support settings (80% BW_Set_ and 90% BW_Set_) on the important physiological training parameters VO_2_, HR and LA [15,16,17]. In accordance with previous findings on conventional treadmills (CON) and LBPPT, respectively, it was hypothesized, that (1) an increase in incline leads to higher VO_2_, HR and LA values, and (2) increasing body weight support is associated with a decrease in VO_2_, HR and LA.

## 2. Materials and Methods

### 2.1. Subjects

Ten well-trained runners (30.2 ± 3.4 years; VO_2max_ = 60.4 ± 4.2 mL kg^−1^ min^−1^, Table 1) took part in the study. The participants were asked to avoid high-intensity and high-volume training sessions 48 h prior to the test and were informed in detail about the study design. Furthermore, the study was approved by the Ethics Committee of the Humboldt University Berlin and is in accordance with the Declaration of Helsinki. All participants provided written consent to participate in the study.

### 2.2. Protocol and Test Design

The participants completed four testing days in two weeks, at the same time of the day (±30 min), with the same individual running shoes to avoid any footwear-related effects [18,19] and with a break between the tests of at least 48 h. The first day, the participants performed an incremental test on a regular treadmill h/p/cosmos saturn^®^ 250/100 (100% BW_Set_; h/p/cosmos sports and medical GmbH, Nußdorf-Traunstein, Germany) to determine VO_2max_ and velocity at VO_2max_ (vVO_2max_). The initial running speed was set to 8 km h^−1^ and was increased by 2 km h^−1^ after each completed stage until volitional exhaustion. Stage duration was 3 min; between the stages there was a rest of 30 s. The second, third and fourth day, the participants started with a standardized warm-up of 10 min at 50% vVO_2max_. Subsequently, they ran a 6-min submaximal trial (twice 3 min, with a break of 30 s in between) at 70% vVO_2max_ on the LBPPT AlterG^®^ Anti-Gravity Treadmill^®^ Pro 200 Plus (AlterG^®^, Fremont, CA, USA). Body weight was set to 80% BW_Set_ and 90% BW_Set_, while incline was altered between 0%, 2% and 7%, in randomized order, resulting in 6 trials of 6 min for each subject. Before each testing day, body weight and height were measured (Seca Vogel and Halke Hamburg 910, seca GmbH and Co. KG, Hamburg, Germany). Oxygen consumption was recorded using a stationary system with breath-by-breath analysis (Quark CPET, COSMED, Pavona di Albano, Italy). HR data were acquired using the HRM Run^TM^ system (Garmin Ltd., Canton Schaffhausen, Switzerland). Before each trial, between the stages and directly after exhaustion of the VO_2max_ test, a sample of 20 μL of arterialized capillary blood was taken from the earlobe, solubilized in a 1000 µL hemolysate solution and analysed using the SUPER GL ambulance system (Dr. Müller Gerätebau GmbH, Freital, Germany). Between the stages and after each test, rating of perceived exertion (RPE) was routinely analysed on the basis of the Borg RPE Scale (6–20) [20].

### 2.3. Data Analysis

VO_2_ data processing was completed via the software OMNIA 1.6 (COSMED, Pavona di Albano, Italy). According to current research, VO_2max_ was determined as the highest value averaged over 30 s [21]. As suggested, by Billat and Koralsztein [22], vVO_2max_ was defined as the lowest running velocity maintained for at least one minute that elicited VO_2max_. If a participant reached VO_2max_ but did not maintain one minute of running, the velocity of the previous stage was used as vVO_2max_, as introduced by Kuipers et al. [23]. If, in turn, the running velocity was maintained for at least one minute (i.e., one third of stage duration), vVO_2max_ was considered to be the running velocity of the previous stage plus one third of the increase between the two stages. Analogously, in case of two minutes maintained (i.e., two thirds of stage duration), vVO_2max_ was approximated by be the running velocity of the previous stage plus two thirds of the increase between the two stages. Any other acquired submaximal VO_2_ and HR data were averaged over 30 s, including the last 30-s-value of each stage for statistical analysis [21].

### 2.4. Statistics

To detect an effect of incline and BW_Set_ on VO_2_, HR and LA data, two-way repeated measures analysis of variance (ANOVA) with Bonferroni posthoc tests were performed. To check sphericity, Mauchly’s test was applied. If the assumption of sphericity was violated, the Greenhouse–Geisser correction was used. Data were processed with IBM SPSS Statistics 23 (IBM, Armonk, NY, USA) and Microsoft Excel 2016 (Microsoft Corporation, Redmond, WA, USA). Results are presented as mean ± standard deviation. Standard level of significance was set to *p* = 0.05, effect sizes were evaluated on the basis of eta squared (*η*^2^). In addition, 95% confidence intervals (CI) were calculated.

## 3. Results

### 3.1. Oxygen Consumption (VO_2_)

The effect of incline and BW_Set_ on VO_2_ is presented in Figure 1. Incline showed a significant main effect (*p* < 0.001, *η*^2^ = 0.881), as did BW_Set_ (*p* < 0.001, *η*^2^ = 0.954). A significant interaction effect between incline and BW_Set_ was not found (*p* = 0.429, *η*^2^ = 0.076). 

Post hoc comparisons revealed a significant mean difference of 6.8 ± 0.8 mL kg^−1^ min^−1^ between 0% incline and 7% incline (*p* < 0.001, CI: 4.3–9.3), and of 5.4 ± 0.8 mL kg^−1^ min^−1^ between 2% and 7% incline (*p* < 0.001, CI: 3.2–7.6), respectively. No significant difference was observed for 0% vs. 2% incline (*p* = 0.117). Furthermore, between 80% BW_Set_ and 90% BW_Set_, VO_2_ differed by 3.3 ± 0.2 mL kg^−1^ min^−1^ (*p* < 0.001, CI: 2.7–3.8).

### 3.2. Heart Rate (HR)

The effect of incline and BW_Set_ on HR is presented in Figure 2. A significant main effect was revealed for both incline (*p* < 0.001, *η*^2^ = 0.657) and BW_Set_ (*p* < 0.001, *η*^2^ = 0.876). Furthermore, a significant interaction effect between incline and BW_Set_ was not found (*p* = 0.766, *η*^2^ = 0.029).

Post hoc comparisons showed a significant mean difference of 12 ± 2 bpm between 0% and 7% incline (*p* < 0.05, CI: 6–19), and of 10 ± 2 bpm between 2% and 7% incline (*p* < 0.05, CI: 3–16), respectively. No significant difference was found for 0% vs. 2% incline (*p* = 0.793). Additionally, post hoc comparisons for 80% BW_Set_ vs. 90% BW_Set_ showed mean differences of 6 ± 1 bpm (*p* < 0.001, CI: 4–8).

### 3.3. Blood Lactate Concentration (LA)

Figure 3 depicts the effect of incline and BW_Set_ on LA. Significant main effects could be observed for incline (*p* < 0.05, *η*^2^ = 0.397) and BW_Set_ (*p* < 0.001, *η*^2^ = 0.783). A significant interaction effect between incline and BW_Set_ was not found (*p* = 0.069, *η*^2^ = 0.257). Despite the clear main effect, the post hoc comparisons showed no difference between 0% and 2% incline (*p* = 1.000), 0% incline and 7% incline (*p* = 0.086) nor between 2% and 7% incline (*p* = 0.067). However, post hoc comparisons for 80% BW_Set_ vs. 90% BW_Set_ showed mean differences of 0.10 ± 0.02 mmol L^−1^ (*p* < 0.001, CI: 0.06–0.15).

## 4. Discussion

The present study is, to the best of our knowledge, the first to examine how uphill running on a LBPPT effects VO_2_, HR and LA. Confirming our initial hypothesis, it could be observed that an increase in incline on LBPPT generally leads to higher VO_2_ and HR. However, no significant difference could be determined for LA. Furthermore, it was found that an increase in body weight support (from 90% BW_Set_ to 80% BW_Set_) results in a decrease in VO_2_, HR and LA. Neither for VO_2_, HR nor for LA interaction effects between incline and BW_Set_ were found.

A detailed analysis of the collected data suggests that for uphill running, a certain amount of incline is necessary to induce physiological changes. This is shown by the fact that between 0% vs. 2% incline a significant difference in VO_2_, HR and LA could not be observed. However, we noticed VO_2_ changes for 0% vs. 7% incline (≈+20%) and for 2% vs. 7% incline (≈+15%), respectively. Changes in HR amounted to ≈+9% (0% vs. 7% incline) and ≈+7% (2% vs. 7% incline). Regarding LA values, however, despite a significant main effect, no significant pointwise results could be obtained for the post hoc comparisons. Probably, this is due to a type I error [24], so that we rely on the results of the post hoc test at this point [24] and omit further interpretation. As for body weight support, there was a VO_2_-increase for 90% BW_Set_ vs. 80% BW_Set_, which corresponds to relative reductions of 9–10%. The HR-values showed a relative increase of 4–5%. For LA, there was a significant difference of 0.10 ± 0.02 mmol L^−1^.This corresponds to relative increase of ≈9%.

Due to the fact that no further data regarding the effect of uphill running on LBPPTs are available, a comparison with corresponding results is currently not feasible. To circumvent this dilemma, existing data on conventional treadmills (CON) may be taken as sole reference at this point: Padulo et al. [25] showed a VO_2_-decrease of 10.2 mL kg^−1^ min^−1^ for 0% vs. 7% incline and 4.8 mL kg^−1^ min^−1^ for 2% vs. 7% incline, which represents a relative increase of 18.7% and 8.0%, respectively. The data of the present study are in accordance with those results, although absolute increases turn out to be slightly different (6.8 ± 0.8 mL kg^−1^ min^−1^ for 0% vs. 7% incline and 5.4 ± 0.8 mL kg^−1^ min^−1^ for 2% vs. 7% incline) while percentage increases are slightly higher (≈20% and ≈15%). Considering the appreciably higher VO_2max_ (76.3 ± 2.6 mL kg^−1^ min^−1^) of the participants taking part in the study of Padulo et al. [25], it may be hypothesized that subjects’ performance level may have had an influence on those results. 

Participants of the present study and those of Padulo et al. ran at 70% vVO_2max_, corresponding to mean velocities of 12.6 km h^−1^ in the present study and 15.0 km h^−1^ in the study of Padulo et al. [25]. Biomechanical factors such as ground contact time and stride rate as well as pattern of muscle innervation may have influenced physiological reactions [4]. Even though changes in VO_2_ associated with incline are similar, there is a substantial difference: while on LBPPTs we observed no change between 0% and 2% incline, Padulo et al. found a significant increase of 5.4 mL kg^−1^ min^−1^ (+9.9%) [25]. Thus, it appears that on the used LBPPT at 70% vVO_2max_, a 2% incline does not yet suffice to induce significant physiological changes in VO_2_, as opposed to running on a CON, where significant changes are present [25]. With the data at hand, it remains unclear whether this “non-adaptation” on the LBPPT for 2% incline observed at the common low-intensity training stimulus of 70% vVO_2max_ [16] will also persist at other training intensities, e.g., during a high-intensity training (≥88% VO_2max_) [16]. This question should be clarified in future research.

Furthermore, HR data from the present study are in accordance with the results of Padulo et al., reporting an increase from 148 ± 12 bpm to 155 ± 12 bpm and 170 ± 12 bpm at 0%, 2% and 7% incline, respectively [25]. We observed HR values of 139 ± 6 bpm (0% incline), 142 ± 5 bpm (2% incline) and 152 ± 4 bpm (7% incline). This indicates that the increases are higher on a CON that on a LBPPT: the relative increments were about 15% (0% to 7% incline) and 10% (2% to 7% incline), respectively. Slightly lower increases could be observed at LBPPT with ≈9% (0% to 7% incline) and ≈7% (2% to 7% incline). In analogy to VO_2_ changes, no differences could be observed for 0% vs. 2% incline on the LBPPT, while on the CON there was an increase of ≈5% [25].

Concerning the LA values, distinct differences between LBPPT and CON were observed. On the LBPPT we could not detect any differences in LA with regard to the inclines. In comparison, for CON values ranging from 2.5 ± 0.9 mmol L^−1^ to 3.4 ± 1.5 mmol L^−1^ and 9.5 ± 2.3 mmol L^−1^ were reported for 0%, 2% and 7% incline, respectively, representing relative increases of +280% and +179% [25]. Those considerably higher LA values in the study of Padulo et al. raise the question whether the initial load at 0% incline in that study still corresponded to a standard low-intensity training stimulus of zone 1—which should then be below 1.5 mmol L^−1^ [16]. Due to the increasing incline and the associated additional physiological demand, there are inevitably increases, which are also related to the exponential response of LA above the threshold [26,27]. In view of the values of the present study (1.15 ± 0.21 mmol L^−1^ at 0% incline), it becomes clear that the training stimulus in the present study was generally lower. Probably because of that, no exponential increases were reached. Moreover, it is shown that for LA, not even an incline of 7% (on the LBPPT at 70% vVO_2max_) is sufficient to achieve physiological–metabolic adaptations.

Regarding the factor BW_Set_, our VO_2_ results of mean decreases of 2.7–3.8 mL kg^−1^ min^−1^ for 90% BW_Set_ vs. 80% BW_Set_ are in accordance with previous studies. Farina et al., e.g., concluded that each decrease in BW_Set_ by 10% is associated with a reduction of approximately 3.4 mL kg^−1^ min^−1^ [4]. This is very similar to our mean value of 3.3 ± 0.2 mL kg^−1^ min^−1^. Other authors report larger decreases: McNeill et al. showed in elite distance runners a decrease of 13.3 mL kg^−1^ min^−1^ at 3.84 m s^−1^ (≈13.8 km h^−1^) and 18.5 mL kg^−1^ min^−1^ at 5.36 m s^−1^ (≈19.3 km h^−1^) for a 20% decrease in BW_Set_, respectively [1]. Those authors hypothesized that highly trained runners benefit “better” from BW_Set_ than well-trained runners [2]. On the other hand, running velocity and accordingly physiological demand may have an impact [1,2]. Furthermore, data from the present study suggest that BW_Set_ leads to a decrease in HR. The decreases from 90% BW_Set_ to 80% BW_Set_ amounted to 4–8 bpm, while other studies reported mean decreases of 15 bpm for running at 10–18 km h^−1^ [2] and of 20 bpm at 13.8 km h^−1^ [1] for 20% decrease in BW_Set_, respectively. Additionally, the significant decrease in LA values for 90% BW_Set_ vs. 80% BW_Set_ of the present study is qualitatively in line with previous results, although significantly larger reductions of 1.0 mmol L^−1^ were reported for 100% BW_Set_ vs. 80% BW_Set_ (10–18 km h^−1^) [2]. In the study of Fleckenstein et al., LA values stayed unaffected by the LBPPT at lower running velocities, while clear changes were found for faster running velocities [2]. Regarding previous research, those results suggest that changes in VO_2_, HR and LA can vary substantially, depending on the tested participants and their performance level. Therefore, future studies on inter-individual responses on the LBPPT may prove helpful for practitioners.

As initially pointed out, elite runners tend to train on the LBPPT with increasing usage frequency [1] to reduce musculoskeletal loading [4], e.g., ground reaction forces and peak tibial accelerations, while maintaining a preset physiological stimulus. From a practical point of view, it seems reasonable to bring those different aspects together: based on an exemplary low-intensity training stimulus, running at 80% BW_Set_ is associated with a VO_2_ reduction of 6.8 mL kg^−1^ min^−1^ [4]. Adding an incline of 7%, that BW_set_-induced reduction could almost ideally be counterbalanced by an opposite increase in VO_2_ by 6.8 ± 0.8 mL kg^−1^ min^−1^ due to incline. Thus, physiological demand would almost stay identical. From a biomechanical perspective, there is evidence that BW_Set_ implies lower ground reaction forces [3,4,10,12,28], while peak tibial accelerations are unaffected by the LBPPT [29,30]. Although no biomechanical analyses were pursued in the current study, the existing investigations on CON suggest that peak impact forces as well as the peak tibial accelerations are reduced during uphill running [31], while only the horizontal propulsive peak forces increase [32]. Hence, by combining physiological and biomechanical perspectives and factors, it could be hypothesized that a 7% incline (at 80% BW_Set_) might be useful to keep VO_2_- and HR-load equal, while keeping biomechanical stress as low as possible. With such an incline, no increase in running velocity would be required (to re-increase VO_2_), preventing peak impact forces and peak tibial accelerations from rising to potentially harmful levels. Future studies should pursue this question and elucidate the interplay between physiological and biomechanical factors in inclined LBPPT running.

## 5. Conclusions

To the best of our knowledge, the present study is the first to investigate how uphill running on a LBPPT effects the three commonly used parameters for monitoring training—VO_2_, HR and LA. It was found that a 2% incline at a given low-intensity stimulus of 70% vVO_2max_ is not sufficient to modify VO_2_, HR and LA values. However, a 7% incline increases VO_2_ and HR significantly. It can be assumed that the increased physiological load at 7% incline is approximately equivalent to the reduced physiological load of 20% body weight reduction (80% BW_Set_) from previous studies [4]. Therefore, from a practical training perspective, such an incline on a LBPPT seems effective to keep physiological load high without increasing running velocity, which is associated with increasing biomechanical load. Future studies should evaluate both physiological and biomechanical variables in the same uphill running study design. In addition, it should be investigated whether the influence of incline is also given to the same extent in forms of high-intensity training.

## Figures and Tables

**Figure 1 sports-09-00051-f001:**
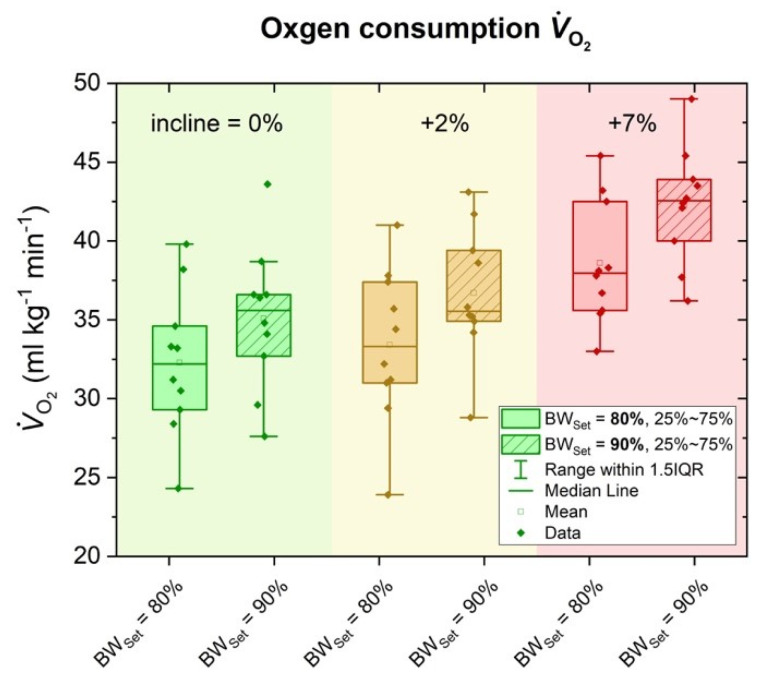
Oxygen consumption VO_2_ in terms of 0% (green), 2% (yellow) and 7% (red) incline and compared for running on 80% BW_Set_ and 90% BW_Set_.

**Figure 2 sports-09-00051-f002:**
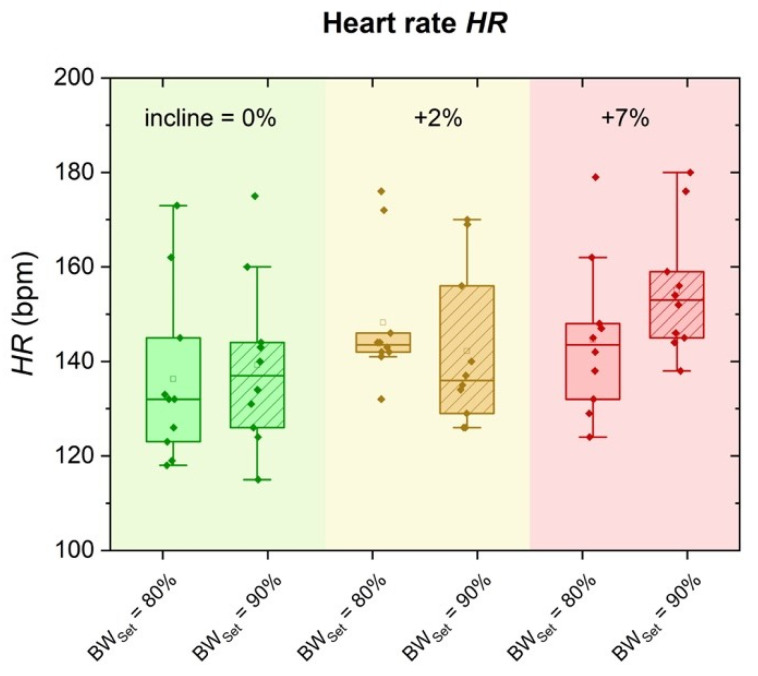
Heart rate HR in terms of 0% (green), 2% (yellow) and 7% (red) incline and compared for running on 80% BW_Set_ and 90% BW_Set_.

**Figure 3 sports-09-00051-f003:**
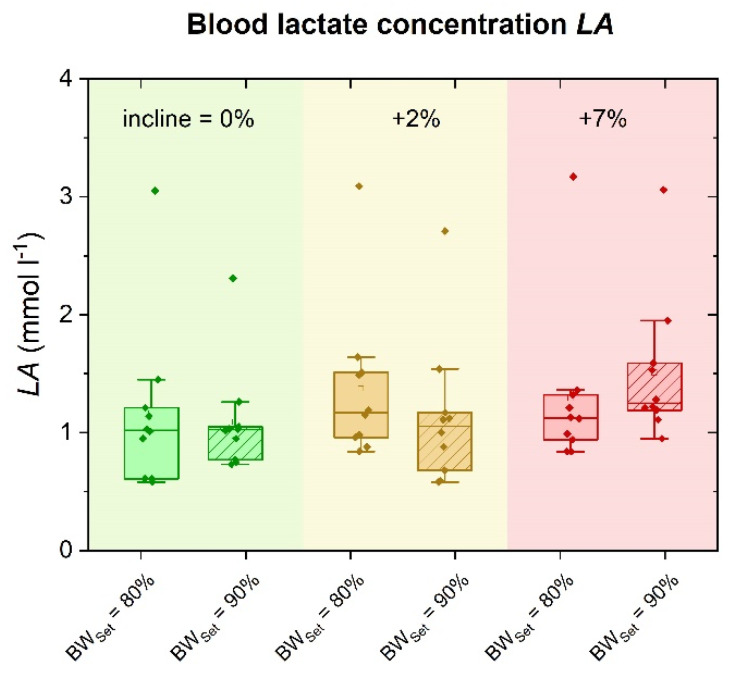
Blood lactate concentration LA in terms of 0% (green), 2% (yellow) and 7% (red) incline and compared for running on 80% BW_Set_ and 90% BW_Set_.

**Table 1 sports-09-00051-t001:** General characteristics of participants included in the study. VO_2_: oxygen consumption; HR: heart rate.

Measure	Male (*n* = 10) (min–max)
Age (years)	30.2 ± 3.4 (25.7–37.1)
Body mass (kg)	73.1 ± 6.1 (63.7–82.2)
Body height (cm)	178 ± 6 (166–185)
VO_2max_ (mL min^−1^ kg^−1^)	60.4 ± 4.2 (54.9–68.4)
vVO_2max_ (km h^−1^)	18.0 ± 1.6 (16.0–20.0)
HR_max_ (beats min^−1^)	193 ± 8 (184–209)

## Data Availability

The data presented in this study are available on request from the corresponding author.

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
