# Peer review of "Effect of Uphill Running on VO2, Heart Rate and Lactate Accumulation on Lower Body Positive Pressure Treadmills"

_sports, 2021, doi:10.3390/sports9040051_

Round 1

Reviewer 1 Report

Abstract:

  • Line 15 - … commonly used in the context of competitive sports. – I am not certain the use of LBPPT are common yet but ‘common’ is subjective. It’s an acceptable opening sentence.
  • Line 16 - Opening the second sentence with ‘Nevertheless, the effects…. are unclear.’ Contradicts the ‘common’ in the first. Perhaps combine both, “LBPPT are becoming more common as part of sports conditioning although the requisite physiological parameters are unclear.” Just an idea for consideration.
  • Lactate accumulation may be easier to read through out if abbreviated LA vs La.
  • Why inclines of 0, 2, and 7 vs, 0, 3, 6, etc. with even break points? No need to redo but perhaps simply say why those values were selected.
  • No mention of why we use an LBPPT in abstract (decrease impacts to lower extremity, specifically after injury or to increase volume and avoid injury)
  • No statistical values in abstract, simply mention of significant changes, no word on what or how this was arrived at.

Introduction:

  • Purpose and hypothesis clearly mentioned.
  • Subjects explained
  • Line 74-75 – The ‘Written informed consent…’ sentence reads oddly. Consider revising. “All participants provided written consent to participate”, etc….
  • Protocol clear and reproducible
  • Data analysis clear
  • Results – again, would be nice to see rational for the 0, 2, and 7% incline. It is possible that a difference may have been present at 3 or 6 and not necessitate a full jump from 2 to 7.

Discussion:

  • Lines 175 and 177 use 1) and 2) to reference the hypothesis. Consider fi the sentence would read clearer without the digits simply referencing the hypothesis.
  • Line 183 add ‘significant’ between ‘…incline a significant difference…’
  • It may be author or journal preference to some degree but much of what is in the discussion could be moved to the introduction to set a clearer stage on the rational for the study. A significant amount of new material was introduced in the discussion. Specifically musculoskeletal loading.
  • As noted, a weakness of the study is that no ground reaction forces where assessed. That the 7% incline may have altered those to a point where a recommendation to run at 90% BW and 7 degrees incline may be counterproductive to lower body stress.

Reviewer 2 Report

The manuscript examines the effects of different uphill running conditions at 80% and 90% body weight support, at 0%, 2% and 7% incline. Specifically, the study aimed to analyse the effect on oxygen consumption, heart rate and blood lactate accumulation. The authors have concluded that 7% incline at 80% and 90% body weight support may be used to keep Oxygen consumption, heart rate and partly blood lactate accumulation load unchanged as compared to unsupported running, while biomechanical stress is substantially reduced.

The study is simple and original concerning the effects of uphill running with body weight support on physiological parameters. Despite the authors not having assessed directly unsupported running condition and biomechanical factors the study is well conducted and written.

  • Authors must justify why 80% and 90% of BWSet with a contol unsupported running condition was not compared?

  • In my opinion this paper can be more interesting for rehabilitation and return in competition field than to be used like method training for competitive athletes. I propose that the authors should also give a direction towards the introduction and the discussion on this field.

  • The statistical analysis is not appropriate. The statistical analysis does not include the factor BWSet and examined 80% body weight support and 90% body weight support separately.

To validate the purpose of this study and defend the difference between 80% body weight support and 90% body weight support in the discussion authors must take into consideration the factor BWSet, with the three incline levels in the same time and not separately.

Authors wrote in line 120 « If a significant main effect could be observed, a one-way repeated measures ANOVA (for incline) and a paired sample t-test (for BWSet) were applied to test for simple effects ». In this case significant main effects or interactions must be followed by a Bonferoni post hoc test.

Authors must realize the new statistical analysis, the new figures accordingly, and the new results of the statistical analysis.

Round 2

Reviewer 1 Report

Thank for making the adjustments.

Reviewer 2 Report

Thanks for the corrections

I aggre with the corrections but the Figure 1 of the paper is missing

Cordially